# Burden of chronic obstructive pulmonary disease in adults aged 70 years and older, 1990–2021: Findings from the Global Burden of Disease Study 2021

**Kaifang Meng**[1☯], **Xu Chen**[2☯], **Zhishang Chen**[3], **Jing Xu**[4,5]*

1 Department of Respiratory and Critical Care Medicine, Nanjing Drum Tower Hospital Clinical College of Nanjing Medical University, Nanjing, Jiangsu, People's Republic of China, 2 School of Public Health and Healthcare Management, Gannan Medical University, Ganzhou, Jiangxi, People's Republic of China, 3 Department of Intensive Care Unit, Xuzhou Center Hospital, The Affiliated Xuzhou Hospital of Medical College of Southeast University, Xuzhou, Jiangsu, People's Republic of China, 4 Department of Respiratory and Critical Care Medicine, The Second People's Hospital of Fuyang City, Fuyang, Anhui, People's Republic of China, 5 Department of Respiratory and Critical Care Medicine, Fuyang Infection Disease Clinical College of Anhui Medical University, Fuyang, Anhui, People's Republic of China

☯ These authors contributed equally to this work.

* 12764543@qq.com

**Data Availability Statement:** The data analyzed in this study are all publicly available at Global Health Data Exchange (https://vizhub.healthdata.org/gbd-results/).

## Abstract

### Background

Life expectancy at age 70 has continued to rise globally over the past 30 years. However, a comprehensive assessment of the burden of COPD in older adults is lacking. We aimed to estimate the burden of COPD and its attributable risk factors among adults aged ≥70 years.

### Methods

Data on the prevalence, incidence, deaths, disability-adjusted life years (DALYs), and risk factors of COPD among adults aged ≥70 years from 1990 to 2021 across 204 countries and territories, were sourced from the Global Burden of Disease Study 2021. Estimated annual percentage change (EAPC) was used to illustrate temporal trends at global and regional levels from 1990 to 2021.

### Results

In 2021, the global numbers of prevalent and incident COPD cases among older adults were 99.7 and 7.4 million, increasing by 162.2% and 157.4% from 1990. The prevalence and incidence rates increased from 18823.5 (95% uncertainty interval (UI) 16324.4–21208.4) to 20165.6 (17703.8–22549.4) and 1429.0 (1224.2–1613.0) to 1502.7 (1309.0–1677.9) per 100,000 population (EAPC 0.31, 95% CI 0.28–0.33; 0.17, 95% CI 0.16–0.19). The global numbers of COPD-associated deaths and DALYs in 2021 reached 2.9 and 45.4 million, increasing by 70.7% and 70.0% from 2019, while the corresponding rates declined (both EAPC <0). The highest prevalence and the largest increase in incidence rate occurred in high sociodemographic index (SDI) regions, while the largest increase in death and DALY

**Funding:** This study was supported by the Scientific Research Project of Fuyang City, Anhui Province (FY2020xg01). There was no additional external funding received for this study.

**Competing interests:** The authors have declared that no competing interests exist.

**Abbreviations:** CI, confidence intervals; COPD, chronic obstructive pulmonary disease; DALYs, disability-adjusted life-years; EAPCs, estimated annual percentage changes; FEV$_1$, the first second of expiration; FVC, forced vital capacity; GBD, global burden of diseases; SDI, socio-demographic index; UI, uncertainty interval; YLD, years lived with disability; YLL, years of life lost.

rates occurred in the low SDI regions. The United States had the highest prevalence rates in 2021, while Iran had the largest increase. From 1990 to 2021, the death rates attributable to ambient ozone pollution-related COPD in older adults have risen, particularly in low and low-middle SDI regions.

## Conclusion

COPD in older adults has progressively become a global health challenge with rising prevalence and incidence rates. Although the death and DALY rates attributed to COPD have globally decreased in older adults, the absolute counts are rapidly increasing. The inequalities across different regions and countries underscore a multi-faceted approach to COPD management in older adults.

## Introduction

Chronic obstructive pulmonary disease (COPD) is a heterogeneous condition characterized by persistent, usually progressive airflow limitation and airway inflammation, along with systemic manifestations [1, 2]. Various risk factors and comorbidities are linked to the disease, such as ageing, smoking, occupational exposures, nonoptimal temperatures, low socioeconomic status, cardiovascular disease, diabetes, and obstructive sleep apnea [3–6]. Some comorbidities may occur independently of COPD, while others may be causally related, either due to shared risk factors or by one disease increasing the severity or risk of the other, ultimately leading to a significant impact on survival [7, 8]. Recent evidence from the global burden of disease (GBD) study [9] indicated that although the age-standardized rates of prevalence, mortality, and disability-adjusted life years (DALYs) declined overall in the past three decades, significant inequalities persist among different population groups.

Since 1990, the global population of older adults has been steadily increasing, accompanied by a decline in all-cause death rates for both males and females. Evidence from GBD showed that life expectancy at 70 has continued to rise globally over the past 30 years [10]. The projected ageing demographics are correlated with the escalating burden and extended duration of non-communicable diseases [11]. Recent studies on the burden of COPD have been global, regional, and national, reporting the epidemiological trends mainly for the entire age range of the population [12–14]. Although it is recognized that the trends in older adults may differ, a comprehensive assessment of the burden of COPD among adults aged ≥70 remains scarce [10]. The GBD study is a key resource for understanding the epidemiological landscape of diverse diseases [15]. Utilizing the latest 2021 GBD study, we acquired COPD-specific data from 1990–2021 in the age ≥70 years group, offering a comprehensive breakdown of prevalence, incidence, deaths, DALYs of COPD, and its attributable risk factors by age, sex, region, countries, and social development index (SDI). This study offers additional evidence supporting the implementation of precise policies and strategies to manage the future increased burden of COPD.

## Materials and methods

### Overview and data sources

The GBD 2021 offers a comprehensive epidemiological evaluation of 371 diseases and injuries, and 88 risk factors, covering 21 GBD regions, and 204 countries/territories from 1990 to 2021

[15]. Prevalence and incidence for COPD were estimated using DisMod-MR 2.1 (Disease Modelling Meta-Regression; version 2.1), a Bayesian meta-regression tool that generates consistent estimates by sex, location, year, and age group [15]. Cause of death estimates were derived using the Cause of Death Ensemble model (CODEm), which combines multiple statistical models and covariate combinations to predict deaths by location, age, sex, and year. The methodology for data acquisition, processing, and analysis in the GBD 2021 study has been thoroughly detailed in prior publications [16, 17]. COPD is defined according to the Global Initiative for Chronic Obstructive Lung Disease (GOLD), where the ratio of forced expiratory volume in the first second of expiration to forced vital capacity ($FEV_1/FVC$) <0.70 after bronchodilation. The codes of COPD in the International Classification of Diseases 10th Revision are J41-J44, with corresponding codes in the 9th Revision being 491–492 and 496 [18]. The burden estimation for COPD was based on an extensive analysis of 203 primary sources, including literature reviews, GOLD proportion data, hospital claims data, and insights from the Burden of Obstructive Lung Disease (BOLD) study [15]. For this study, age- and location-specific prevalence, deaths, and DALYs along with 95% uncertainty intervals (UIs) for COPD from 1990–2021 were downloaded from the Global Health Data Exchange (GHDx) (https://vizhub.healthdata.org/gbd-results/). To profile the age distribution of COPD burden among adults aged ≥70 years, patients were categorized into six groups: 70–74, 75–79, 80–84, 85–89, 90–94, and 95+ years. This study followed the Guidelines for Accurate and Transparent Health Estimates Reporting (GATHER) [19].

## The socio-demographic index

In the GBD study, the SDI is utilized as a comprehensive metric to depict the development of each region alongside health outcomes. It represents the geometric mean of the total fertility rate among individuals under 25 years, the mean education level of those above 15 years, and the per capita income distribution [20]. These factors are normalized on a 0 to 1 scale, with higher values corresponding to higher development relevant to health, and vice versa indicating lower development. Using the 2021 SDI values, 204 countries were classified into 5 groups: high, high-middle, middle, low-middle, and low quintiles.

## Ethic

All data used in this study were from publicly available databases and did not contain any individually identifiable information; therefore, no ethics approval or consent to participate are needed.

## Statistical analysis

In 2021, the numbers and rates of prevalence, death, and DALYs were reported by age, sex, region, country, and SDI. DALYs quantify the health impact of a disease by measuring the years of healthy life lost due to the condition, combining years of life lost (YLL) from premature death and years lived with disability (YLD) into a single metric: DALYs = YLL + YLD. Per the GBD algorithm, 95% uncertainty intervals (UIs) for all estimates were obtained by averaging data from 500 draws with replacement, with the interval limits set by the 2.5th and 97.5th ranked values among all draws [21]. In addition, the mean estimated annual percentage changes (EAPCs), and its 95% confidence interval (CI) were used to determine temporal trends through a linear regression model [22]. An increasing trend is implied when both the EAPC and the lower limit of the 95% CI >0, while a decreasing trend is suggested when both the EAPC and the upper limit of the 95% CI <0. To determine the relationship between rates and SDI, we used locally estimated scatterplot smoothing (LOESS) regression to draw a

predicted value line. Additionally, risk factors and trends from 1990–2021 for COPD in the age ≥70 years group were assessed. All calculations were performed using R Studio version 4.1.3.

## Results

### Global trends

In 2021, the global number of prevalent and incident cases of COPD among adults aged ≥70 years was 99.7 and 7.4 million, increasing by 162.2% and 157.4% from 1990. The corresponding prevalence and incidence rates increased from 18823.5 (95% UI 16324.4–21208.4) to 20165.6 (17703.8–22549.4) and 1429.0 (1224.2–1613.0) to 1502.7 (1309.0–1677.9) per 100,000 population, with the EAPC of 0.31 (95% CI 0.28–0.33) and 0.17 (95% CI: 0.16–0.19) (Table 1 and S1 Table in S1 File). In 2021, the prevalent and incident cases in both sexes decreased with

**Table 1. Prevalence and EAPC of chronic obstructive pulmonary disease among adults aged ≥70 years at the global and regional levels, from 1990 to 2021.**

| | Prevalence (95% UI) | | | | EAPCs (95% CI) |
|---|---|---|---|---|---|
| | No in 1990, thousands | Rate in 1990 (per 100,000) | No in 2021, thousands | Rate in 2021 (per 100,000) | |
| Global | 38024.8 (32976.4–42842.4) | 18823.5 (16324.4–21208.4) | 99692.1 (87121.9–111476.9) | 20165.6 (17703.8–22549.4) | 0.31 (0.28–0.33) |
| **SDI** | | | | | |
| High SDI | 14529.9 (12671.3–16375.7) | 21040.5 (18349.0–23713.3) | 31379 (28472.4–34275.3) | 21869.8 (19844.0–23888.4) | 0.25 (0.18–0.31) |
| High-middle SDI | 9493.6 (8167.4–10817.8) | 18447.7 (15870.6–21020.9) | 22740.9 (19690.5–25752.0) | 19370.1 (16771.9–21935.0) | 0.25 (0.19–0.31) |
| Middle SDI | 7916.6 (6831.0–8919.7) | 17324.8 (14949.0–19520.0) | 27552.3 (23447.3–31360.5) | 19541.5 (16630.1–22242.5) | 0.45 (0.42–0.49) |
| Low-middle SDI | 4770 (4153.3–5361.8) | 18189.3 (15837.8–20445.9) | 14392.7 (12536.0–16009.7) | 20533 (17884.2–22839.9) | 0.44 (0.42–0.46) |
| Low SDI | 1276.9 (1106.1–1443.9) | 13682.7 (11852.3–15472.0) | 3542.8 (3093.0–3969.9) | 16152.3 (14101.6–18099.4) | 0.57 (0.54–0.59) |
| **Regions** | | | | | |
| Andean Latin America | 119 (98.8–141.4) | 11657.8 (9683.4–13861.0) | 473.2 (396.6–547.5) | 14425.3 (12091.0–16690.6) | 0.84 (0.78–0.90) |
| Australasia | 248 (208.5–279.1) | 17017.6 (14304.7–19153.5) | 519 (442.3–605.8) | 14245 (12140.2–16626.6) | -0.51 (-0.64–0.39) |
| Caribbean | 162.2 (136–187.1) | 10985.1 (9215.4–12677.8) | 473.9 (413.6–531.5) | 14811.1 (12928.5–16612.4) | 1 (0.90–1.10) |
| Central Asia | 371.2 (313.5–428.8) | 16473.9 (13911.6–19030.1) | 607.6 (520.2–690.0) | 17876.3 (15304.6–20325.6) | 0.4 (0.19–0.61) |
| Central Europe | 1338.7 (1131.2–1552.6) | 16947.9 (14320.4–19655.1) | 2896.6 (2535.0–3252.5) | 19512.9 (17076.8–21910.8) | 0.63 (0.56–0.71) |
| Central Latin America | 633.6 (541.4–716.4) | 15685.4 (13402.5–17732.9) | 2705.4 (2367.7–3044.6) | 19803.3 (17331.2–22286.5) | 0.75 (0.67–0.82) |
| Central Sub-Saharan Africa | 82.7 (67.7–98.4) | 10201.2 (8355.3–12142.2) | 239.3 (196.0–284.0) | 12690 (10394.5–15061.8) | 0.69 (0.68–0.70) |
| East Asia | 7679.5 (6680.1–8574) | 19734.6 (17166.3–22033.5) | 24984 (21327.5–28638.0) | 20216.9 (17258.1–23173.7) | 0.18 (0.12–0.24) |
| Eastern Europe | 2625.7 (2167.1–3069.8) | 17331.6 (14304.6–20262.6) | 3263.9 (2756.7–3769.9) | 15279.7 (12905.2–17648.5) | -0.43 (-0.60–0.27) |
| Eastern Sub-Saharan Africa | 274.3 (227.4–326.3) | 8806.6 (7300.4–10476.5) | 689.3 (571.8–807.2) | 10003.6 (8299.0–11715.6) | 0.31 (0.22–0.39) |
| High-income Asia Pacific | 1652.9 (1363.7–1948.3) | 14685.1 (12115.4–17309.3) | 5184.3 (4396.3–6075.5) | 14837.9 (12582.8–17388.9) | 0.16 (0.09–0.23) |
| High-income North America | 5735.9 (5065.7–6401.3) | 24700.4 (21814.3–27565.5) | 12332.4 (11524.6–13019.9) | 28475.8 (26610.5–30063.5) | 0.63 (0.42–0.85) |
| North Africa and Middle East | 974.4 (819.7–1141.6) | 13493 (11351.0–15807.8) | 3776.4 (3233.6–4321.9) | 18571.9 (15902.6–21254.3) | 1.14 (1.10–1.18) |
| Oceania | 18.1 (15.7–20.3) | 17384.1 (15106.9–19572.9) | 46.2 (38.8–52.5) | 16740 (14420.9–19041.9) | -0.13 (-0.18–0.09) |
| South Asia | 4915.7 (4301.3–5510.1) | 20928.8 (18312.8–23459.4) | 16837.5 (14723.7–18653.0) | 22997.4 (20110.4–25477.2) | 0.36 (0.33–0.39) |
| Southeast Asia | 1550.4 (1285.5–1817.6) | 14201.8 (11775.0–16649.1) | 4621.3 (3852.9–5380.1) | 15364.4 (12809.9–17887.2) | 0.23 (0.21–0.25) |
| Southern Latin America | 306.9 (250.3–371.1) | 11645.9 (9497.6–14083.3) | 677.1 (579.1–775.4) | 12317 (10533.5–14105.6) | 0.3 (0.17–0.43) |
| Southern Sub-Saharan Africa | 183.7 (153.8–214) | 14151.3 (11850.8–16488.1) | 398 (330.5–463.7) | 14925.5 (12392.9–17385.9) | 0.18 (0.14–0.22) |
| Tropical Latin America | 824.3 (681.7–957.9) | 18878.7 (15612.9–21939.5) | 3008.1 (2533.2–3472.2) | 20943.1 (17636.6–24174.2) | 0.23 (0.19–0.27) |
| Western Europe | 7986 (6984.7–8938.4) | 21384 (18702.9–23934.3) | 15099.3 (13502.0–16823.0) | 22948.5 (20520.8–25568.3) | 0.33 (0.30–0.36) |
| Western Sub-Saharan Africa | 341.6 (284–406.2) | 8499.3 (7067.4–10106.8) | 859.4 (715.2–1009.5) | 10544.8 (8775.6–12387.1) | 0.75 (0.70–0.80) |

SDI, Sociodemographic Index; EAPC, estimated annual percentage change; UI, uncertainty interval; CI, confidence interval.

increased age, while the rates showed an upward trend (Fig 1A and 1B). Additionally, the number of prevalent cases in females was higher than in males across 6 age groups, while the incidence rates in males were higher than in females across 6 age groups.

In 2021, the global number of COPD-associated deaths among adults aged ≥70 years was 2.85 million, increasing by 70.7% from 1990. The corresponding death rate decreased from 827.3 (95% UI 740.3–889.7) to 577.2 (513.9–636.3) per 100,000 population, with the EAPC of -1.33 (95% CI -1.40–1.25) (S2 Table in S1 File). In 2021, the number of COPD-associated deaths was highest in the 80–84 age group in both sexes, after which the numbers decreased with increasing age (Fig 1C). The death rate attributed to COPD peaked at 90–94 years for males and 95+ years for females. The global COPD death rate in older adults was higher in males than in females across six age groups (Fig 1C).

In 2021, the global number of COPD-associated DALYs among adults aged ≥70 years was 45.4 million, increasing by 70.0% from 1990. The corresponding DALY rate decreased from 13467.0 (95% UI 12220.2–14429.4) to 9189.5 (8377.6–9950.4) per 100,000 population, with the EAPC of -1.41 (95% CI-1.48–1.34) (S3 Table in S1 File). When stratified by age and sex, the trends in both the number and rate of DALYs were similar to those of deaths (Fig 1D).

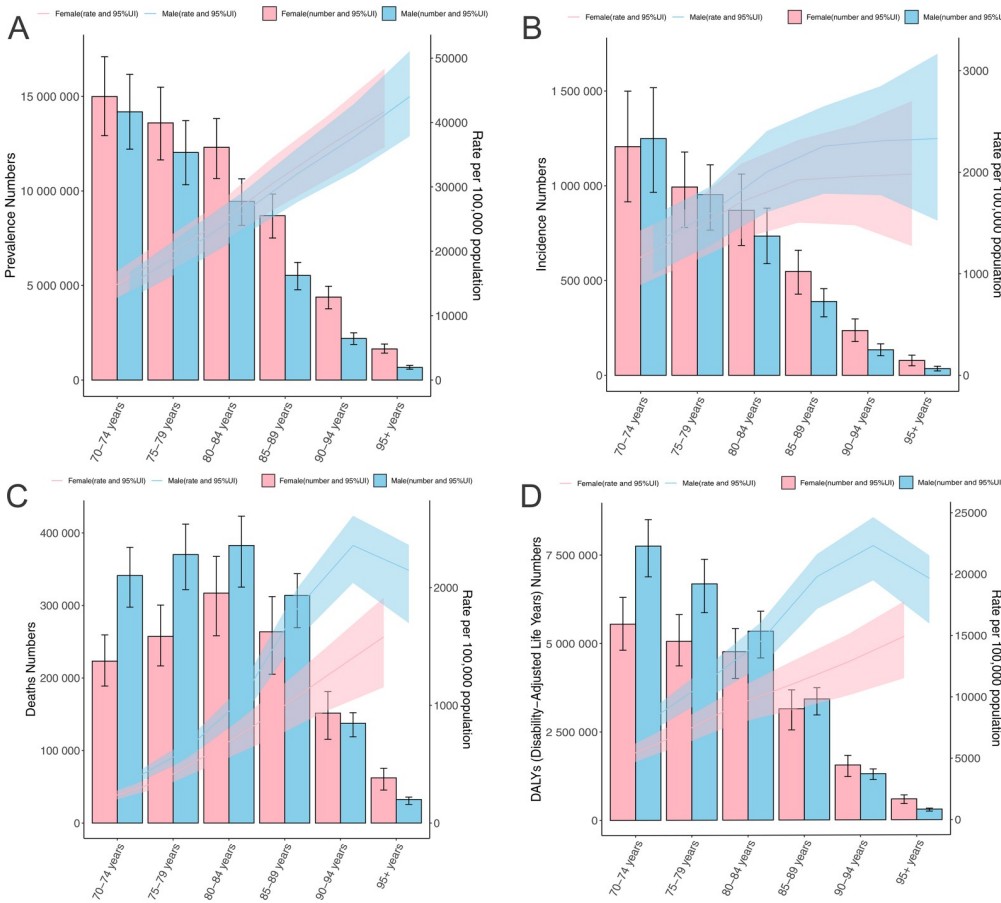

**Fig 1. The global numbers of prevalent cases (A), incident cases (B), deaths (C), DALYs (D), and corresponding rates of chronic obstructive pulmonary disease per 100,000 population in older adults, by age and sex in 2021.** Shaded areas are 95% uncertainty intervals. DALY, disability-adjusted-life-years.

## SDI regional trends

The prevalent and incident cases and rates of COPD among adults aged ≥70 years increased across five SDI regions from 1990 to 2021 (all EAPC >0) (Table 1 and S1 Table in S1 File). In 2021, the high SDI regions had the most prevalent cases of 31.4 (95% UI 28.5–34.3) million and rate of 21869.8 (95% UI 19844.0–23888.4) per 100,000 population, while the low SDI region had the least prevalent cases of 3.5 (95% UI 3.1–4.0) million and rate of 16152.3 (95% UI 14101.6–18099.4) per 100,000 population (Table 1 and S1 Table in S1 File). The greatest increase in the prevalence rates occurred in the low SDI region, while the greatest rise in the incidence rates occurred in the high SDI (Table 1, Fig 2A and 2B, S1 Table in S1 File).

The number of deaths attributed to COPD among adults aged ≥70 years increased across all SDI regions from 1990 to 2021, with the greatest increase occurring in the low-middle SDI region (171.5%) (S2 Table in S1 File). The low SDI and low-middle SDI regions exhibited

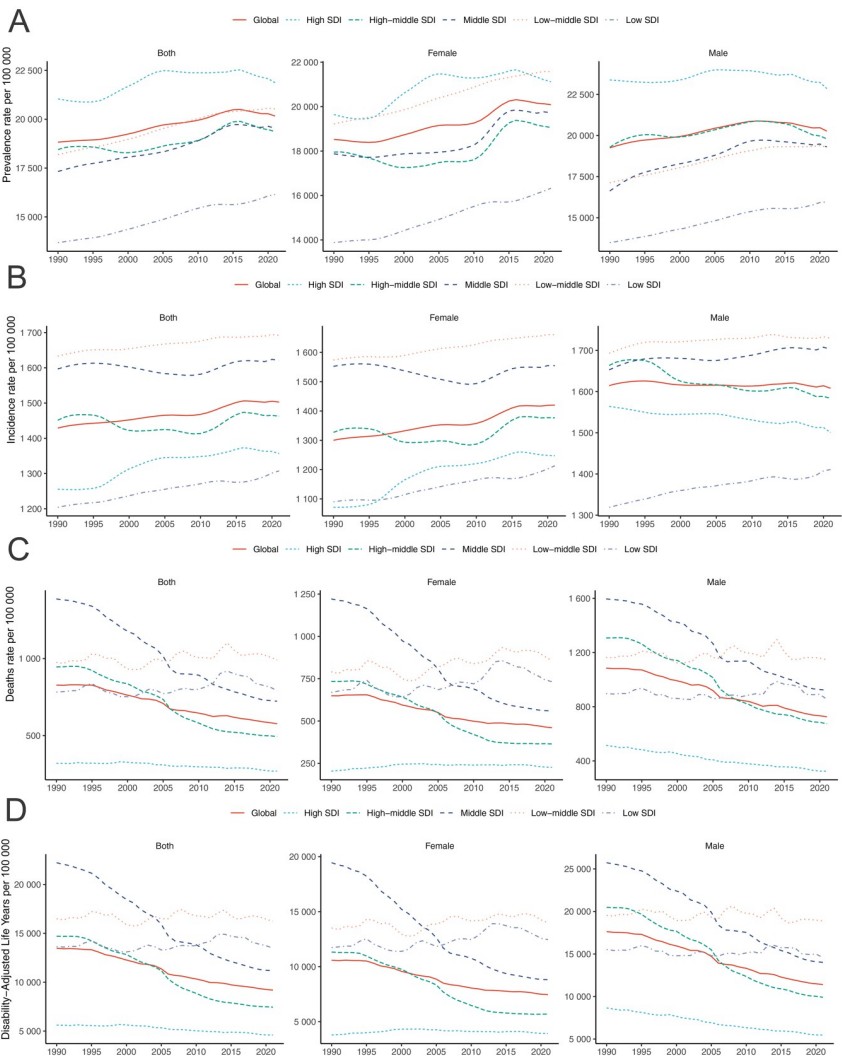

**Fig 2. Trends of prevalence (A), incidence (B) deaths (C), and DALYs (D) rates of chronic obstructive pulmonary disease per 100,000 population among adults aged ≥70 years by sex and sociodemographic index, from 1990 to 2021.** DALY, disability-adjusted life-years.

increased COPD-associated death rates (EACP 0.31, 95% CI 0.13–0.49; 0.20, 95% CI 0.07–0.33), while the rates of the remaining 3 SDI regions were declined. In 2021, the highest death rate was observed in low-middle SDI regions (990.8, 95% UI 884.3–1096.9), while the lowest death rate occurred in the high SDI regions (268.6, 95% UI 229.2–289.6) (Fig 2C, S2 Table in S1 File). From 1990 to 2021, the rates of COPD-associated deaths in high SDI regions increased for females but declined for males (Fig 2C).

The number of DALYs caused by COPD among adults aged ≥70 years increased across five SDI regions from 1990 to 2021, with the greatest increase occurring in the low-middle SDI region (163.1%) (S3 Table in S1 File). Only low-SDI regions exhibited increased COPD-associated DALY rates (EACP 0.17, 95% CI 0.04–0.29). In 2021, the DALYs rates were highest in low-middle SDI regions (16230.2, 95% UI 14695.5–17783.7), while the lowest rate of COPD-associated DALYs occurred in the high SDI regions (4593.3, 95% UI 4117.7–4905.7) (Fig 2D, S3 Table in S1 File). When stratified by sexes, the trends of DALY rates were similar to deaths (Fig 2D).

### Geographic regional trends

In 2021, East Asia, South Asia, and Western Europe had the most prevalent and incident cases of COPD among adults aged ≥70 years, while high-income North America and South Asia had the highest prevalence and incidence rates, respectively (Table 1 and S1 Table in S1 File). From 1990 to 2021, North Africa and Middle East had the largest increase in both the prevalence and incidence rates (EAPC 1.14, 95% CI 1.10–1.18; 1.18, 95% CI 1.14–1.21) (Table 1, S1 Table and S1, S2 Figs in S1 File). In 7 regionals (eg, East Asia, and South Asia), the prevalence rates of COPD in females were higher than in males (S3 Fig in S1 File). In comparison, the incidence rates in females were lower than in males, except for South Asia (S4 Fig in S1 File) In addition, we found a positive association between the prevalence rate and SDI among adults aged ≥70 years from 1990 to 2021, with 7 regions (eg, East Asia, South Asia) exceeding expected prevalence rates based on their SDI (Fig 3). The association between the incidence rate and SDI was shown in S5 Fig in S1 File.

East Asia (1.13 million), South Asia (0.86 million), and Western Europe (0.18 million) had the most death cases of COPD among adults aged ≥70 years in 2021, while the highest death rates occurred in Oceania (1222.6) (S2 Table in S1 File). From 1990 to 2021, most geographic regions exhibited declining trends of death rates except for high-income North America and the Caribbean (EAPC: 0.79, 95% CI 0.47–1.11; 0.41, 95% CI 0.21–0.61) (S2 Table and S6 Fig in S1 File). The death rates of males were higher than females across 21 geographic regions (S7 Fig in S1 File). In addition, we found a reversed V-shaped association between the SDI and COPD death rates among adults aged ≥70 years from 1990 to 2021. East Asia, South Asia, and Oceania exhibited higher death rates than expected based on their SDI levels (S8 Fig in S1 File).

In 2021, East Asia (16.73 million), South Asia (14.21 million), and Western Europe (2.77 million) had the most DALYs cases of COPD among adults aged ≥70 years, while the highest DALYs rates occurred in Oceania (19732.7) (S3 Table in S1 File). From 1990 to 2021, only high-income North America and the Caribbean (EAPC >0) exhibited increased trends of DALY rates (S3 Table and S9 Fig in S1 File). The DALY rates of males were higher than females across 21 geographic regions (S10 Fig in S1 File). Similarly, a reversed V-shaped association between the SDI and DALY rate was observed, and 4 regions (eg, high–income North America and East Asia) had higher DALY rates than expected based on their SDI levels (S11 Fig in S1 File).

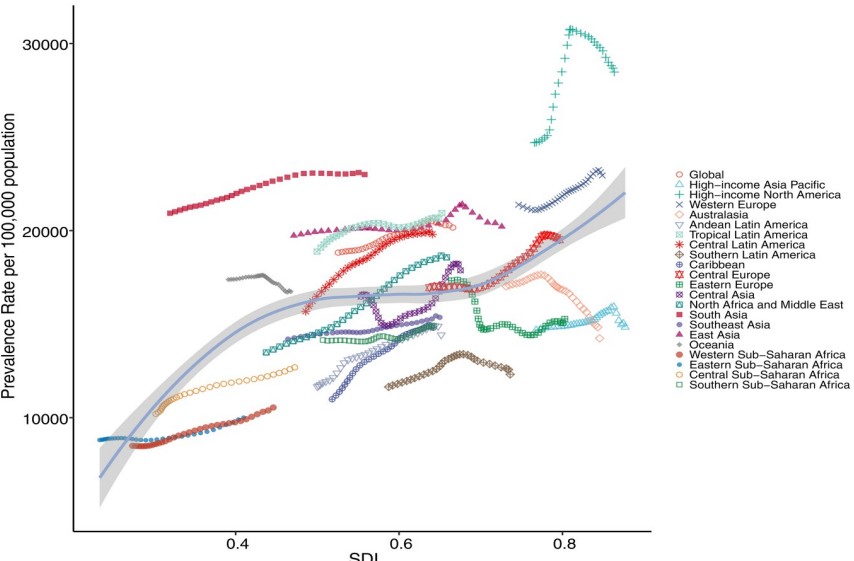

**Fig 3. The association between the sociodemographic index and the prevalence rates of chronic obstructive pulmonary disease per 100,000 population among adults aged ≥70 years across 21 regions, from 1990 to 2021.** For each region, the thirty-two points from left to right represent the estimates for each year from 1990 to 2021. The solid line represents the expected values derived from the sociodemographic index and disease rates in all locations.

## National trends

In 2021, China, India, and the United States had the most prevalent and incident cases of COPD among adults aged ≥70 years (S4 and S5 Tables in S1 File). The United States (29634.8), United Kingdom (28537.8), and Sweden (27750.8) had the highest point prevalence rates (Fig 4A, S4 Table in S1 File), while Iran (EAPC 1.91, 95% CI 1.83–2.00), Lebanon (EAPC: 1.79, 95% CI 1.71–1.87), Estonia (EAPC: 1.49, 95% CI 1.35–1.62) had the largest increase in prevalence rates (Fig 5A, S4 Table in S1 File). As for incidence rates, Nepal had the highest point prevalence rate (2242.7), while Iran (EAPC 2.02, 95% CI 1.93–2.11) had the largest increase (Figs 4B and 5B, S5 Table in S1 File). At the country level in 2021, the prevalence and incidence rates of COPD increased with increasing SDI (r = 0.4811, P <0.001; r = 0.342, P <0.001) (S12, S13 Figs in S1 File).

In 2021, China (1.10 million), India (0.75 million), and the United States (0.15 million) had the most deaths cases attributed to COPD among adults aged ≥70 years (S6 Table in S1 File). Nepal (1608.9), Papua New Guinea (1597.8), and North Korea (1317.0) had the highest death rates, with Kuwait (41.9), Montenegro (67.9), and Latvia (73.3) having the lowest estimates (Fig 4C, S6 Table in S1 File). Georgia (EAPC 3.23, 95% CI 2.45–4.02), Norway (EAPC 2.91, 95% CI 2.31–3.52), and Greece (EAPC: 1.64, 95% CI 1.01–2.27) had the largest increase in death rates, whereas Belarus (EAPC: -7.57, 95% CI -8.12–7.02), Ukraine (EAPC -6.44 95% CI -6.79–6.09), and Singapore (EAPC -5.80, 95% CI -6.05–5.74) had the largest decreases (Fig 5C, S6 Table in S1 File). At the country level in 2021, the death rates of COPD gradually increased with SDI before 0.4 but rapidly declined after SDI >0.5 (S14 Fig in S1 File).

In 2021, China (16.28 million), India (12.30 million), and the United States (2.78 million) had the most DALY numbers of COPD among adults aged ≥70 years (S7 Table in S1 File). Nepal (26125.5), Papua New Guinea (25697.1), and India (20644.7) had the highest DALYs rates, with Kuwait (1461.3), Singapore (1514.7), and United States Virgin Islands (1561.5) having the lowest estimates (Fig 4D, S7 Table in S1 File). Georgia (EAPC 2.42 95% CI 1.90–2.95),

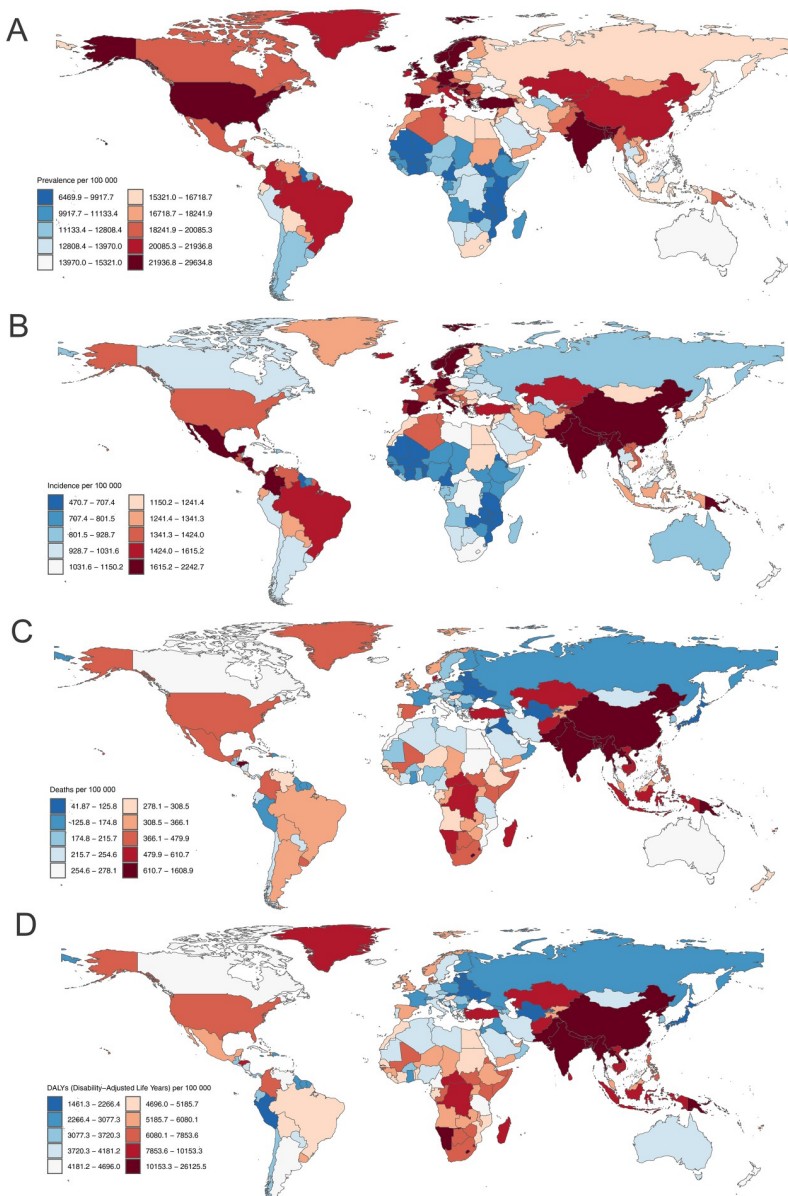

**Fig 4. Map showing prevalence (A), incidence (B), deaths (C), and DALYs (D) rates of chronic obstructive pulmonary disease per 100,000 population among adults aged ≥70 years in 2021, by country.**

Norway (EAPC 2.15, 95% CI 1.68–2.62), and United Arab Emirates (EAPC 1.52, 95% CI 0.83–2.22) had the largest increase in DALYs rates, whereas Belarus (EAPC -6.17, 95% CI -6.64–5.69), Ukraine (EAPC -5.60, 95% CI -5.91–5.29), and Singapore (EAPC -5.45, 95% CI -5.61–5.30) had the largest decreases (Fig 5D, S7 Table in S1 File). At the country level in 2021, the trends of DALY rates with SDI were similar to the results of death rates (S15 Fig in S1 File).

## Risk factors for COPD in older adults

In 2021, 7 risk factors for COPD were identified in the GBD comparative risk assessment framework. Globally, COPD related to particulate matter pollution contributed to the highest

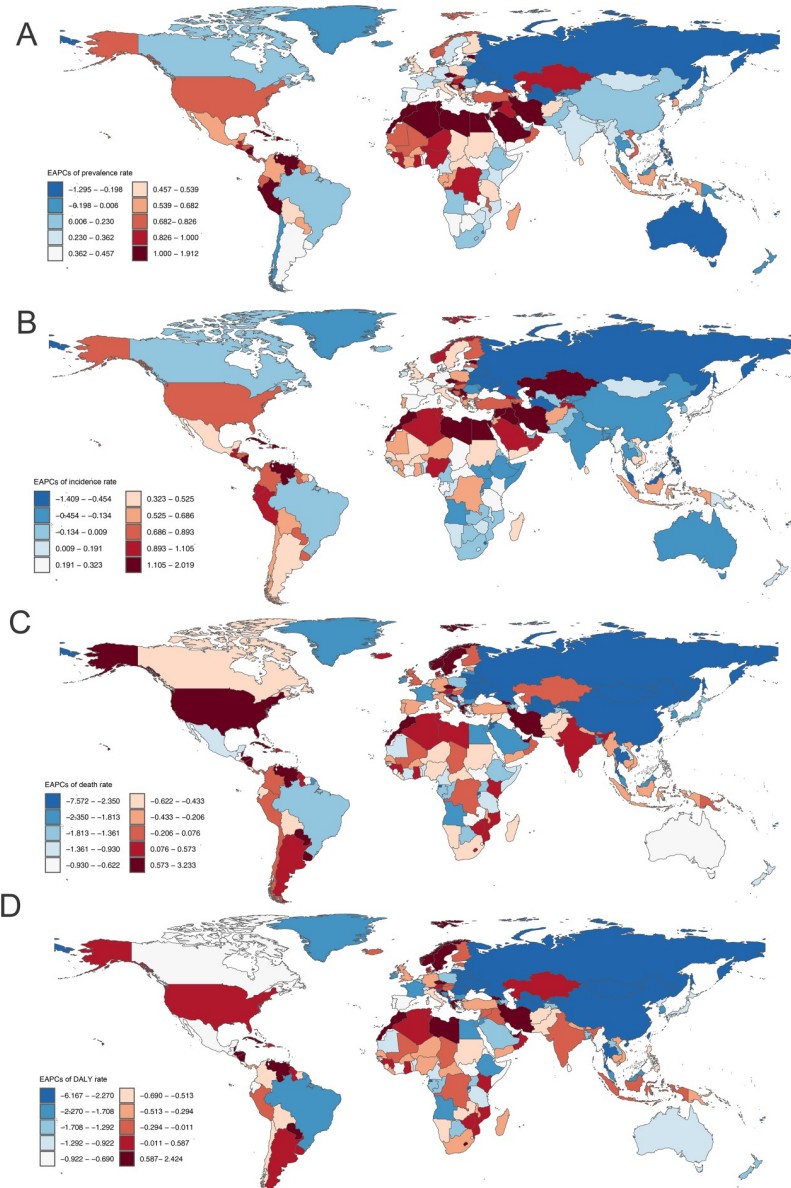

**Fig 5. Map showing estimated annual percentage changes of prevalence (A), incidence (B), deaths (C), and DALYs (D) rates of chronic obstructive pulmonary disease among adults aged ≥70 years in 2021, by country.** DALY, disability-adjusted life-years.

death rate per 100,000 among adults aged ≥70 years, followed by smoking, and occupational particulate matter, gases, and fumes (S16 Fig in S1 File). However, in high SDI regions, smoking and low temperature were ranked 1st and 2nd contributors to the COPD-related death rate. In high−middle SDI regions, the 1st and 2nd contributors to the COPD-related death rate were smoking and particulate matter pollution. From 1990 to 2021, death rates attributed to particulate matter pollution- and smoking-related COPD have decreased, while the death rates attributed to ambient ozone pollution-related COPD among adults aged ≥70 years have risen, particularly in low and low-middle SDI regions (Fig 6).

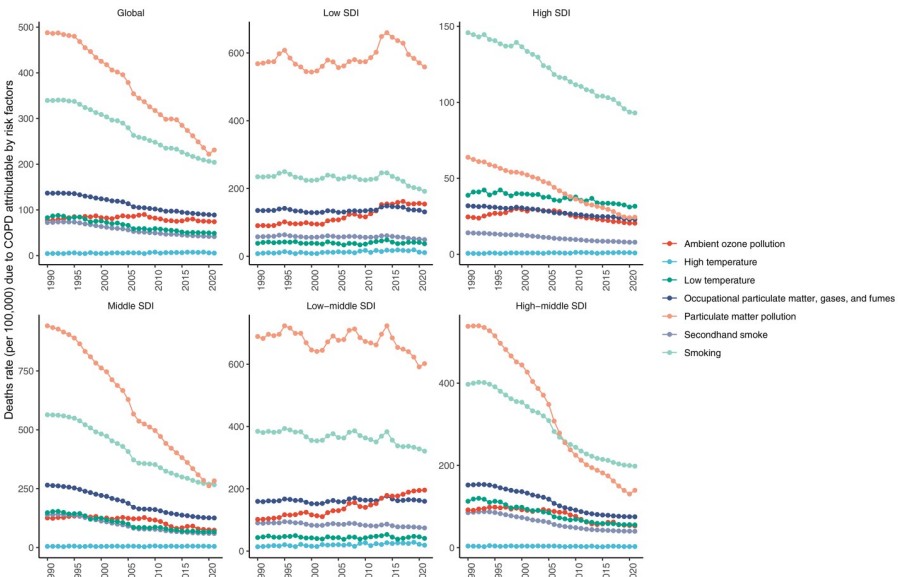

**Fig 6. Trends of death rates due to chronic obstructive pulmonary disease attributable to each risk factor among adults aged ≥70 years in 5 sociodemographic index regions, from 1990–2021.**

## Discussion

To our knowledge, this is the first systematic evaluation of the global burden and risk factors of elderly COPD across regions and countries from 1990 to 2021. COPD in older adults has progressively become a global health challenge with rising prevalence and incidence across 5 SDI regions. The greatest prevalence rates occurred in the high SDI regions, especially in the United States, the United Kingdom, and Sweden. Although the death and DALYa rates have decreased globally, they are still increasing in some regions, especially in low SDI regions. The absolute counts of death and DALYs are increasing, likely due to population growth, ageing demographics, and increased life expectancy. The primary risk factors of COPD among adults aged ≥70 years have declined in the past three decades, while the death rates attributable to ambient ozone pollution-related COPD have risen, particularly in low and low-middle SDI regions.

A nationally representative sample of Chinese adults aged ≥20 revealed a spirometry-based COPD prevalence of 8.4%, with rates escalating with age, peaking at 35.5% in the ≥70 age group [23]. In addition, nationally representative results of US adults reported a prevalence rate of 9.6% in ≥70 years [24]. In our results, the COPD prevalence rates among adults aged ≥70 in China and the US were 20.3% and 29.6%, which is inconsistent with these studies. Various factors contributing to this disparity include the study year, diagnostic criteria, participant numbers, and methodologies for COPD prevalence reporting [26]. Hence, the findings of individual studies on COPD burden cannot be compared with the results from the GBD. In addition, the COVID-19 pandemic had a substantial impact on COPD globally, which caused a huge disruption to healthcare services, lockdowns, and reduced access to diagnostic and treatment facilities. This may lead to delays in COPD diagnoses and treatments, which in turn resulted in an increase in COPD cases during and after the pandemic [27–29]. Results from 2019 indicated that in the age groups ≥75–79 years, the number of females with COPD surpasses that of males, despite having lower prevalence rates [26]. In contrast, data from 2021 reveals that when aged ≥70–74, both the number of female cases and their prevalence rates

(except for 95+ years) exceed those of males, signaling a growing challenge posed by COPD in elderly women. Evidence from the GBD 2019 showed that the age-standardized prevalence rates of COPD declined over the past 3 decades [9, 26], which was consistent with estimates from the National Health Interview Survey (NHIS) 1999–2011 [30] and the Behavioral Risk Factor Surveillance System (BRFSS) 2011–2021 [25]. Our study concentrated on the aged ≥70 years and found that the prevalence and incidence rates continue to increase, exhibiting similar trends to the previous results [25]. Among the different SDI regions, the high SDI region showed both the highest prevalence and the greatest increase in incidence rates. This trend may be attributed not only to ageing populations and increased life expectancy but also to better access to healthcare, advanced screening, and more active case detection in these regions [31]. In contrast, the low SDI regions, particularly Eastern and Western Sub-Saharan Africa, reported the lowest prevalence and incidence rates, which are likely due to higher rates of underdiagnosis, limited medical care, and a lack of diagnostic resources in these areas [9, 32]. Consistently, a positive association was found between the prevalence rates of elderly COPD and SDI, which differed from the findings of a previous study on the general population [26]. Therefore, in high SDI regions, where the ageing population contributes to increased COPD prevalence, policies should prioritize geriatric care, chronic disease management, and preventive healthcare. In low SDI regions, policies must focus on strengthening primary healthcare systems and increasing the availability of basic diagnostic tools for COPD, such as spirometers [31]. In addition, across all geographic regions, high-income North America and South Asia had the highest point prevalence and incidence rates in 2021, respectively, while North Africa and Middle East had the largest increase. These regional inequalities highlight the need for a multifaceted approach to COPD management in older adults.

In 2021, the death rates attributed to COPD in older adults were dramatically higher compared with the previous results on the general population in 2019 (577.2 vs 42.5 per 100,000 population) [26]. In elderly individuals, reduced lung function due to ageing, impaired lung tissue repair, and underlying inflammation physiologically elevate the risk of mortality [33, 34]. Furthermore, COPD is linked to worsened health status and a higher burden of comorbidities [6], which coupled with the inherent comorbidities of old age, amplifies the COPD death rates in the elderly population. In addition, COVID-19 itself contributed to COPD-related complications, exacerbations, and mortality [35], either directly or indirectly, which might lead to a temporary rise in the observed burden of elderly COPD. As for the global trends from 1990 to 2021, the death and DALY rates attributed to elderly COPD decreased over the past 3 decades, consistent with trends previously reported in the GBD for the general population [9, 26, 36, 37]. The decreased burden of COPD may be attributed to various recent strategies, including widespread tobacco control measures, environmental pollution management, enhanced access to disease-modifying treatments and management of comorbidities, and advancements in screening tools [26, 38, 39]. We found a lack of concordance between the point prevalence and death rates due to COPD in country rankings. For instance, despite the United States having the highest point prevalence of COPD among the elderly, it was ranked 52nd in terms of death rates. This discrepancy is probably due to advancements in diagnostic methods and improved management of COPD and related conditions. In contrast, even though Nepal had the highest death rate, it was placed 16th in point prevalence rates, possibly due to premature mortality.

Previous studies showed that smoking and ambient particulate matter pollution were identified as the 1st and 2nd risk factors of COPD in the entire population globally [4, 14, 26]. This study found that particulate matter pollution was the most important contributor to the death rate attributable to COPD among adults aged ≥70 years, followed by smoking. Particulate matter 2.5 (PM2.5) is the most frequently monitored particulate matter, comprising minuscule

particles that can linger in the air and be breathed into the lungs [26]. Prolonged exposure to PM2.5 can irritate the airway mucosa, causing inflammation, reduced lung function, accelerated ageing, and harm to lung tissues [14, 40, 41]. In recent years, despite several initiatives being implemented to mitigate exposure to particulate matter pollution [42, 43], the COPD burden from ambient particulate matter still rapidly increased in low and low-middle SDI regions [4]. While in adults aged ≥70, our results found that ambient ozone pollution is the fastest-growing risk factor in the low and low-middle SDI regions, emphasizing that ambient ozone pollution is still a challenge among this population.

This research has several strengths. Firstly, we performed comprehensive estimates of levels and trends of COPD burden among adults ≥70 years by analyzing data for 204 countries and territories from 1990 to 2021. Moreover, we utilized tools from GBD 2021, the most recent version of the database known for its robust design, diverse sample size, and advanced statistical methods. Nevertheless, some limitations need to be addressed. First, underdeveloped countries in the GBD may have underestimated cases because of systemic and infrastructural deficiencies [44, 45]. Secondly, GBD results heavily depend on modeled data, which inherently includes estimations of data completeness, healthcare access, and disease prevalence. These estimations may not accurately mirror real-world conditions, potentially introducing systematic biases that impact the precision and dependability of our conclusions [46]. It is crucial to carefully consider these limitations when interpreting our results. Lastly, in the GBD study, conditions are assessed individually to estimate mortality and morbidity. Yet, given the high prevalence of multimorbidity in older populations, this method may potentially result in an overestimation.

## Conclusion

In conclusion, COPD in older adults has progressively become a global health challenge with rising prevalent and incident cases and rates. The rapidly increasing prevalence in lower SDI regions, alongside the persistently high rates in high SDI regions, underscores a multi-faceted approach to COPD management in older adults. Although the death and DALY rates attributed to COPD have globally decreased in older adults, these rates persist at high levels, especially in low-middle SDI regions. Strategies should be implemented to address the escalating death rates attributed to ambient ozone pollution-related COPD, especially in low and low-middle SDI regions.

## Supporting information

**S1 File.**
(DOCX)

## Acknowledgments

We sincerely appreciate the work of the Global Burden of Disease Study 2021 collaborators.

## Author Contributions

**Conceptualization:** Kaifang Meng, Jing Xu.

**Data curation:** Kaifang Meng, Xu Chen.

**Formal analysis:** Xu Chen.

**Funding acquisition:** Jing Xu.

**Methodology:** Zhishang Chen.

**Project administration:** Jing Xu.

**Software:** Zhishang Chen.

**Supervision:** Jing Xu.

**Writing – original draft:** Kaifang Meng, Xu Chen.

**Writing – review & editing:** Jing Xu.

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
