## [Decision Letter · Decision Letter 0]

28 Nov 2024

PONE-D-24-47807Burden of chronic obstructive pulmonary disease in adults aged 70 years and older, 1990-2021: findings from the Global Burden of Disease Study 2021PLOS ONE

Dear Dr. Xu,

Thank you for submitting your manuscript to PLOS ONE. After careful consideration, we feel that it has merit but does not fully meet PLOS ONE’s publication criteria as it currently stands. Therefore, we invite you to submit a revised version of the manuscript that addresses the points raised during the review process.

We look forward to receiving your revised manuscript.

Kind regards,

Aida Fallahzadeh

Guest Editor

PLOS ONE

**Journal Requirements:**

This study was supported by the Scientific Research Project of Fuyang City, Anhui Province (FY2020xg01). 

4. We note that Figures 4 and 5 in your submission contain map images which may be copyrighted. All PLOS content is published under the Creative Commons Attribution License (CC BY 4.0), which means that the manuscript, images, and Supporting Information files will be freely available online, and any third party is permitted to access, download, copy, distribute, and use these materials in any way, even commercially, with proper attribution. For these reasons, we cannot publish previously copyrighted maps or satellite images created using proprietary data, such as Google software (Google Maps, Street View, and Earth). For more information, see our copyright guidelines: http://journals.plos.org/plosone/s/licenses-and-copyright.

We require you to either present written permission from the copyright holder to publish these figures specifically under the CC BY 4.0 license, or remove the figures from your submission:

a. You may seek permission from the original copyright holder of Figures 4 and 5 to publish the content specifically under the CC BY 4.0 license.  

**Additional Editor Comments:**

Dear Authors,

After careful review of your manuscript, we have decided that it requires major revisions before it can be considered for publication in our journal. While your work presents valuable insights and contributes to the field, several key aspects of the study need further clarification and strengthening. These include (but are not limited to) the methodology, data analysis, and presentation of results. We believe that with substantial revision, your manuscript has the potential to make a significant impact, and we look forward to reviewing the updated version. Please ensure that you address all the reviewer comments thoroughly and provide a point-by-point response when resubmitting your revised manuscript.

Best Regards,

Aida Fallahzadeh, MD

Reviewer 1:

Thank you for inviting me to review this paper. I am not convinced about writing a paper for patients >70 when the more general paper about it is already published. Overall, I think that adding the incidence rates is required so the authors can discuss the significance of this health problem based on the data from death/DALY/prevalence.

- Abstract: The method is too brief and more details are required. The authors can check previous GBD studies to get idea for improving this section.

- Abstract: There is no mention of risk factors in the aim of the study.

- The reason for increased prevalence can be attributed to higher life expectancies. Thus, we cannot use it to highlight the problem as the death rate is also decreasing. The authors should mention the trend for incidence rate as well. Also, it is important to discuss the results by its incidence throughout the manuscript.

-line 93: abbreviations needs to be defined.

- There was no mention of COVID-19 in this paper, which needs to be discussed in GBD2021 results, especially in respiratory diseases.

- The authors have not discussed about the policymaking and interventions. The can suggest new ones and also discuss the previous ones that led to reducing the burden of disease.

Reviewer 2: 

The manuscript is well-written and the overall quality of the introduction and discussion is acceptable. However, some minor modifications are suggested:

- The Methods section is too brief. Please provide additional details regarding the models used in the GBD analysis.

- In the GBD 2021 methodology, 500 iterations were used to obtain the 95% uncertainty intervals (UI). Please correct this in the Methods section.

- Please mention that LOESS regression was used to assess the association between SDI and prevalence rates, as presented in Figure 3.

- It is recommended to report the prevalence and DALY numbers for 2021 in the results of the abstract, in addition to the rates already provided.
The manuscript is well-written and the overall quality of the introduction and discussion is acceptable. However, some minor modifications are suggested:

- The Methods section is too brief. Please provide additional details regarding the models used in the GBD analysis.

- In the GBD 2021 methodology, 500 iterations were used to obtain the 95% uncertainty intervals (UI). Please correct this in the Methods section.

- Please mention that LOESS regression was used to assess the association between SDI and prevalence rates, as presented in Figure 3.

Reviewers' comments:

Reviewer's Responses to Questions

**Comments to the Author**

1. Is the manuscript technically sound, and do the data support the conclusions?

Reviewer #1: No

Reviewer #2: Yes

2. Has the statistical analysis been performed appropriately and rigorously? 

Reviewer #1: I Don't Know

Reviewer #2: Yes

3. Have the authors made all data underlying the findings in their manuscript fully available?

Reviewer #1: Yes

Reviewer #2: Yes

4. Is the manuscript presented in an intelligible fashion and written in standard English?

Reviewer #1: Yes

Reviewer #2: Yes

5. Review Comments to the Author

**Reviewer #1:** Thank you for inviting me to review this paper. I am not convinced about writing a paper for patients >70 when the more general paper about it is already published. Overall, I think that adding the incidence rates is required so the authors can discuss the significance of this health problem based on the data from death/DALY/prevalence.

- Abstract: The method is too brief and more details are required. The authors can check previous GBD studies to get idea for improving this section.

- Abstract: There is no mention of risk factors in the aim of the study.

- The reason for increased prevalence can be attributed to higher life expectancies. Thus, we cannot use it to highlight the problem as the death rate is also decreasing. The authors should mention the trend for incidence rate as well. Also, it is important to discuss the results by its incidence throughout the manuscript.

-line 93: abbreviations needs to be defined.

- There was no mention of COVID-19 in this paper, which needs to be discussed in GBD2021 results, especially in respiratory diseases.

- The authors have not discussed about the policymaking and interventions. The can suggest new ones and also discuss the previous ones that led to reducing the burden of disease.

**Reviewer #2: **The manuscript is well-written and the overall quality of the introduction and discussion is acceptable. However, some minor modifications are suggested:

- The Methods section is too brief. Please provide additional details regarding the models used in the GBD analysis.

- In the GBD 2021 methodology, 500 iterations were used to obtain the 95% uncertainty intervals (UI). Please correct this in the Methods section.

- Please mention that LOESS regression was used to assess the association between SDI and prevalence rates, as presented in Figure 3.

- It is recommended to report the prevalence and DALY numbers for 2021 in the results of the abstract, in addition to the rates already provided.

6. PLOS authors have the option to publish the peer review history of their article (what does this mean?). If published, this will include your full peer review and any attached files.

Reviewer #1: No

Reviewer #2: **Yes: **Ali Golestani

---

## [Author Response · Author response to Decision Letter 0]

3 Dec 2024

Dear editor and reviewers:

Thank you very much for all the nice and insightful comments, which dramatically improved the manuscript quality. We have responded to all the comments point-by-point as shown below and have made revisions in the manuscript accordingly. The revisions in the lines of the revised manuscript have been provided. 

Responses to journal requirements:

Reply: Thanks for your comments. Thanks for your comments. I will make sure the manuscript meets PLOS ONE's style requirements, including those for file naming.

This study was supported by the Scientific Research Project of Fuyang City, Anhui Province (FY2020xg01). 

Please provide an amended statement that declares *all* the funding or sources of support (whether external or internal to your organization) received during this study, Please also include the statement “There was no additional external funding received for this study.” in your updated Funding Statement. 

Reply: Thanks for your comments. We have included the amended Funding Statement within our cover letter as follows：

This study was supported by the Scientific Research Project of Fuyang City, Anhui Province (FY2020xg01). There was no additional external funding received for this study.

3. PLOS requires an ORCID iD for the corresponding author in Editorial Manager on papers submitted after December 6th, 2016.

Reply: Thanks for your comments. I appreciate the reminder regarding the ORCID iD requirement for the corresponding author. I will ensure that the ORCID iD is added to the submission in Editorial Manager as per PLOS guidelines.

4. We note that Figures 4 and 5 in your submission contain map images which may be copyrighted. All PLOS content is published under the Creative Commons Attribution License (CC BY 4.0), which means that the manuscript, images, and Supporting Information files will be freely available online, and any third party is permitted to access, download, copy, distribute, and use these materials in any way, even commercially, with proper attribution. For these reasons, we cannot publish previously copyrighted maps or satellite images created using proprietary data, such as Google software (Google Maps, Street View, and Earth).

Reply: Thanks for your thoughtful comments. 

Thank you for your questions regarding the map images in Fig 4 and 5. Please find our responses below:

a) Source of Map Images:

The map images in Fig 4 and 5 were created using the rnaturalearth R package, which provides access to Natural Earth map data. This data is open-source and available at https://github.com/ropensci/rnaturalearth.

b) Copyright Status:

To our knowledge, the map data from Natural Earth provided through the rnaturalearth package is in the public domain and therefore not subject to copyright restrictions. Natural Earth map data is widely used for world mapping and is explicitly designed for open, unrestricted use. We believe no additional copyright permission is required. 

Please let us know if any further clarification is required, or if additional details or documentation are needed to satisfy PLOS ONE’s requirements.

Responses to Reviewer 1:

1.Abstract: The method is too brief and more details are required. The authors can check previous GBD studies to get idea for improving this section.

Reply: Thanks for the reviewer's comments. We agree with the reviewer's opinion and have expanded description of our methodology in the revised manuscript (lines 34-35).

2. Abstract: There is no mention of risk factors in the aim of the study.

Reply: Thanks for your thoughtful consideration. We agree with the reviewer's opinion and have added the description of risk factors to the aim of the study (lines 29-30).

3. The reason for increased prevalence can be attributed to higher life expectancies. Thus, we cannot use it to highlight the problem as the death rate is also decreasing. The authors should mention the trend for incidence rate as well. Also, it is important to discuss the results by its incidence throughout the manuscript.

Reply: Thanks a lot for your professional advice. We agree with your opinion, the incidence of COPD among older adults has also increased over the past 3 decades. We have added the description of incidence (S1 and S5 Tables in File 1; and the corresponding part in Fig1-2, Fig4-5; S2, S4-5, S13 Figs) and discussed the results involving incidence throughout the revised manuscript.

4. -line 93: abbreviations needs to be defined

Reply: Thanks a lot for your thoughtful consideration. We have added the abbreviations of FEV1, FVC in the revised manuscript (lines 469-470).

5. There was no mention of COVID-19 in this paper, which needs to be discussed in GBD2021 results, especially in respiratory diseases.

Reply: Thanks a lot for your professional comment. The COVID-19 pandemic had a substantial impact on COPD globally, which caused a huge disruption to healthcare services, lockdowns, and reduced access to diagnostic and treatment facilities. This may lead to delays in COPD diagnoses and treatments, which in turn resulted in an increase in COPD cases during and after the pandemic. In addition, COVID-19 itself contributed to COPD-related complications, exacerbations, and mortality, either directly or indirectly, which might lead to a temporary rise in the observed burden of elderly COPD. We have added the part of COVID-19 in the discussion in the revised manuscript (lines 348-353; lines 388-391). 

6. The authors have not discussed about the policymaking and interventions. The can suggest new ones and also discuss the previous ones that led to reducing the burden of disease.

Reply: Thanks a lot for your professional comment. We agree with your opinion and added the policymaking and interventions in the part of the discussion.

(a). In high SDI regions, where the aging population contributes to increased COPD prevalence, policies should focus on geriatric care, chronic disease management, and preventive healthcare. In low SDI regions, policies must focus on strengthening primary healthcare systems and increasing the availability of basic diagnostic tools for COPD, such as spirometers. These regional inequalities highlight the need for a multifaceted approach to COPD management in older adults (lines 373-377).

(b). We acknowledged the impact of widespread tobacco control measures, environmental pollution management, enhanced access to disease-modifying treatments, and management of comorbidities in the general population. And on this basis, we emphasized that ambient ozone pollution remains a challenge for this population, especially in low and low-middle SDI regions (lines 394-398; lines 416-418).

Responses to Reviewer 2:

The manuscript is well-written and the overall quality of the introduction and discussion is acceptable. However, some minor modifications are suggested:

1. - The Methods section is too brief. Please provide additional details regarding the models used in the GBD analysis.

Reply: Thanks a lot for your professional advice. We agree with your opinion, and we have added the description of DisMod-MR 2.1 (Prevalence and incidence estimates) and CODEm (Cause of death estimates) in the part of ‘Overview and data sources’ (lines 93-98).

2. - In the GBD 2021 methodology, 500 iterations were used to obtain the 95% uncertainty intervals (UI). Please correct this in the Methods section.

Reply: Thanks a lot for your professional comment, and we have corrected it in the part of ‘Statistical analysis’ (line 137).

3. - Please mention that LOESS regression was used to assess the association between SDI and prevalence rates, as presented in Figure 3.

Reply: Thanks a lot for your professional comment, and we have added the LOESS regression in the part of ‘Statistical analysis’ (lines 143-144).

4. - It is recommended to report the prevalence and DALY numbers for 2021 in the results of the abstract, in addition to the rates already provided. 

Reply: Thanks a lot for your professional comment, and we have added the numbers of COPD in 2021 in the abstract.

We appreciate your warm work earnestly and hope that the revisions will meet the requirements. Once again, we are really grateful for the help from the editorial team and the reviewers. Hope everything is going well. 

Yours sincerely，

Jing Xu.

---

## [Editor Report · Decision Letter 1]

6 Dec 2024

Burden of chronic obstructive pulmonary disease in adults aged 70 years and older, 1990-2021: findings from the Global Burden of Disease Study 2021

PONE-D-24-47807R1

Dear Dr. Xu,

We’re pleased to inform you that your manuscript has been judged scientifically suitable for publication and will be formally accepted for publication once it meets all outstanding technical requirements.

Kind regards,

Aida Fallahzadeh

Guest Editor

PLOS ONE

Additional Editor Comments (optional):

Dear Dr. Xu,

I hope this message finds you well.

I am pleased to inform you that your manuscript titled "Burden of chronic obstructive pulmonary disease in adults aged 70 years and older, 1990-2021: findings from the Global Burden of Disease Study 2021" has been reviewed following your recent revisions. After carefully assessing your responses to the reviewers' comments and the changes made to the manuscript, I am happy to report that the reviewers are satisfied with the revisions you have provided.

As a result, I will be recommending the manuscript for acceptance for publication.

I would like to thank you for your hard work in addressing the reviewers' feedback, and I appreciate your prompt and thoughtful revisions. We will proceed with the final steps of the publication process and keep you updated on the next stages.

Please feel free to reach out if you have any questions or need further assistance.

Congratulations on reaching this stage, and I look forward to seeing your work published.

Best regards,

Aida Fallahzadeh
---

## [Editor Report · Acceptance letter]

11 Dec 2024

PONE-D-24-47807R1 

PLOS ONE

Dear Dr. Xu, 

I'm pleased to inform you that your manuscript has been deemed suitable for publication in PLOS ONE. Congratulations! Your manuscript is now being handed over to our production team.

Kind regards, 

on behalf of

Dr. Aida Fallahzadeh 

Guest Editor

PLOS ONE